# Efficient room temperature catalytic synthesis of alternating conjugated copolymers via C-S bond activation

Zijie Li[1,2], Qinqin Shi [1,2✉], Xiaoying Ma[3], Yawen Li[4], Kaikai Wen[1], Linqing Qin[1], Hao Chen[1], Wei Huang [5], Fengjiao Zhang[3], Yuze Lin [4], Tobin J. Marks [5✉] & Hui Huang [1,2,6,7✉]

Structural defects in conjugated copolymers are severely detrimental to the optoelectronic properties and the performance of the resulting electronic devices fabricated from them. Therefore, the much-desired precision synthesis of conjugated copolymers with highly regular repeat units is important, but presents a significant challenge to synthetic materials chemists. To this end, aryl sulfides are naturally abundant substances and offer unrealized potential in cross-coupling reactions. Here we report an efficient room temperature poly-condensation protocol which implements aryl disulfide C-S activation to produce defect-minimized semiconducting conjugated copolymers with broad scope and applicability. Thus, a broad series of arylstannanes and thioethers are employed via the present protocol to afford copolymers with number-average molecular weights ($M_n$s) of 10.0–45.0 kDa. MALDI and NMR analysis of selected copolymers reveals minimal structural defects. Moreover, the polymer trap density here is smaller and the field effect mobility higher than that in the analogous polymer synthesized through thermal-activation Stille coupling.

[1] College of Materials Science and Opto-Electronic Technology, University of Chinese Academy of Sciences, Beijing 100049, P. R. China. [2] Center of Materials Science and Optoelectronics Engineering, University of Chinese Academy of Sciences, Beijing 100049, P. R. China. [3] School of Chemical Sciences, University of Chinese Academy of Sciences, Beijing 100049, P. R. China. [4] Beijing National Laboratory for Molecular Sciences, CAS Key Laboratory of Organic Solids, Institute of Chemistry, Chinese Academy of Sciences, Beijing 100190, P. R. China. [5] Department of Chemistry and the Materials Research Center, Northwestern University, 2145 Sheridan Road, Evanston, IL 60208, USA. [6] CAS Center for Excellence in Topological Quantum Computation, University of Chinese Academy of Sciences, Beijing 100049, P. R. China. [7] CAS Key Laboratory of Vacuum Physic, University of Chinese Academy of Sciences, Beijing 100049, P. R. China. ✉email: shiqinqin@ucas.ac.cn; t-marks@northwestern.edu; huihuang@ucas.ac.cn

Semiconducting polymers, especially π-conjugated donor-acceptor (D-A) copolymers with defect-free alternating structures, are essential materials for plastic electronics and optoelectronics owing to their attractions of variable/tunable chemical structures, tunable bandgaps and optical absorptions, excellent charge transport mobility, mechanical flexibility, and solution processability for printing[1–3]. However, to synthesize such complex macromolecules, thermally activated cross-couplings, such as Suzuki-Miyaura, Stille, direct arylation polymerization (DArP), etc., have become the standard procedure[4–6]. Nevertheless, the high reaction temperatures of these polymerizations not only consume energy, but can also compromise chemoselectivity by producing inseparable structural defects in the product polymers[7–10]. Up until now, the polymerizations that can be performed at room temperature or lower in short reaction times have primarily utilized methodologies such as controlled/living polymerization or radical polymerization[11,12]. However, the major limitation of these approaches is lack of regioselectivity, hence unsuitability for creating device-quality alternating conjugated polymers[13].

Recently, the significant need for atom-efficient and environmentally benign chemical synthesis has increased, and turned to process energy-efficiency and starting materials and reagents that must conform to these requirements[14]. Organosulfur compounds that are widely available from nature[15–17] often serve as effective electrophiles in transition metal-assisted cross-coupling reactions, such as Liebeskind-Srogl (L-S) coupling in the case of small molecules (Fig. 1a)[18]. Thus, abundant aromatic thioethers offer a sustainable and environmentally friendlier alternative to traditional aryl halide reagents[19]. Furthermore, C-S bonds are well tolerated functional groups in most other catalytic cross-coupling reactions, such as Suzuki, Stille, etc.[20,21], and importantly, the potentially competing homocoupling of C-S bonds has never been reported, implying that inert C-S bonds will avoid or minimize homocoupling by-products[22–24]. Finally, C-S bond activation using unique additives may substantially lower the activation energies of the desired coupling/polymerization reactions[22,24], thereby allowing polymerization reactions to proceed near room temperature. Therefore, developing a universal and efficient catalytic system to realize the cross-coupling polymerization of aryl thioethers or aryl stannanes based on C-S bond activation presents an intriguing opportunity.

Here we report on the scope and mechanism of a room temperature step-wise cross-coupling polymerization process utilizing aromatic thioethers as electrophiles, representing, to our knowledge, the unprecedent cross-coupling polymerization based on C-S activation (Fig. 1b). Competition experiments indicate that transmetalation is the rate-determining step, indicating that the coupling reaction occurs rapidly between electron-donating nucleophiles and aromatic thioethers, affording more rapid polymerization rates and higher molecular masses than conventional thermal Stille couplings. A wide substrate scope of electron-donating and electron-withdrawing substituted nucleophiles demonstrates the generality of this methodology. Moreover, the present defect studies on the polymers synthesized by our method demonstrate relatively clean NMR spectra, small trap densities, and high hole mobilities, in comparison to those obtained by thermally-activated Stille coupling polymerization.

## Results and discussion

**Polymerization optimization and scope.** Monomer BBT-2SMe (Table 1, **E1**), a derivative of natural waste 2-(methylthio)-benzothiazole (MTBT)[25], synthesized using a lithiation-mediated sulfide transfer reaction, was used as the principal test electrophile in the following polycondensations with 2,6-bis(trimethyltin)-4,8-bis(2-octyldodecyloxy)-benzo[1,2-b:3,4-b']dithiophene (**N1**) selected as the nucleophile (Table 1; Supplementary Table 1). Since elevated temperatures ensure solubility of the growing polymer chain, an initial polymerization between **E1** and **N1** using Pd(PPh$_3$)$_4$ as the catalyst and copper(I) thiophene-2-carboxylate (CuTc) as the cocatalyst was performed at 100 °C (Table 1 entry 1) to afford **P1** with $M_n = 14.5$ kDa and Đ (polydispersity index) = 4.82. When the reaction temperature was lowered to 60 °C, the product polymer had a similar molecular mass and polydispersity ($M_n = 14.3$ kDa, Đ = 3.39) (entry 2). Impressively, a similar product in similar yield is obtained at 25 °C ($M_n = 15.7$ kDa, Đ = 3.15) which is unprecedented for Stille-type polymerizations (entry 3). Not surprisingly, the polymerization affords a lower molecular mass product ($M_n = 7.4$ kDa) and yield when the reaction temperature is lowered to 0 °C. The reaction time is also an important factor in energy efficiency, and was screened from 12 (entry 5) to 72 h (entry 6). Both reactions afford polymers with $M_n$, Đ, and yield similar to those in 24 h. Thus, 12–24 h may be a suitable time range for polymer chain growth. During the polymerization in the above entries, an immediate color change from colorless to deep-red indicates a rapid reaction early stage, consistent with a step-wise mechanism[26]. The amount of catalyst/cocatalyst is also critical for reaction efficiency. Thus, lowering either the Pd (entry 7) or Cu (entry 8) loading results in a dramatic fall in product $M_n$ (but not yield). Next the types of Cu cocatalyst were also screened (entries 9–10), which revealed that a copper carboxylate is optimal, in contrast to previous reports on L-S reactions[24,27].

Since the specific identity of the catalyst is the key factor for these polycondensations, other Pd catalysts were screened (Supplementary Table 1, entries 1–2) for further optimization, which showed that Pd(PPh$_3$)$_4$ is the most efficient. In this

### a Classic Stille Cross-Coupling Polymerization via C-X activation

X= Br, I, etc

### b This Work: Cross-Coupling Polymerization via C-S activation

Advantages: energy- and atom-efficient; homocoupling defects minimized.

**Fig. 1 Synthesis of π-conjugated donor–acceptor copolymers. a** Classic Stille cross-coupling polycondensation. **b** The present cross-coupling polymerization via L-S reaction (Liebeskind-Srogl reaction).

**Table 1 Optimization of C-S cleavage polycondensation conditions for the synthesis of semiconducting polymer P1 from E1 + N1[a].**

| Entry | [Pd] Catalyst | [Cu] Cocatalyst | Solvent | T (°C) | Time (h) | $M_n/M_w$ (kDa) | Đ | Yield[b] (%) |
|---|---|---|---|---|---|---|---|---|
| 1 | Pd(PPh$_3$)$_4$ | CuTc | Toluene | 100 | 24 | 14.5/70.0 | 4.82 | 84 |
| 2 | Pd(PPh$_3$)$_4$ | CuTc | Toluene | 60 | 24 | 14.3/48.7 | 3.39 | 81 |
| 3 | Pd(PPh$_3$)$_4$ | CuTc | Toluene | RT | 24 | 15.7/49.5 | 3.15 | 84 |
| 4 | Pd(PPh$_3$)$_4$ | CuTc | Toluene | 0 | 24 | 7.4/23.4 | 3.17 | 24 |
| 5 | Pd(PPh$_3$)$_4$ | CuTc | Toluene | RT | 12 | 15.1/73.3 | 4.85 | 60 |
| 6 | Pd(PPh$_3$)$_4$ | CuTc | Toluene | RT | 72 | 15.9/60.5 | 3.81 | 70 |
| 7[c] | Pd(PPh$_3$)$_4$ | CuTc | Toluene | RT | 24 | 4.7/12.2 | 2.62 | 62[c] |
| 8[d] | Pd(PPh$_3$)$_4$ | CuTc | Toluene | RT | 24 | 3.7/7.9 | 2.10 | 80[d] |
| 9 | Pd(PPh$_3$)$_4$ | CuMeSal | Toluene | RT | 24 | 7.8/20.9 | 2.67 | 52 |
| 10 | Pd(PPh$_3$)$_4$ | CuI | Toluene | RT | 24 | - | - | 0 |

Reaction conditions: [a]Pd catalyst (10 mol%), Cu cocatalyst (5 equiv.), **E1** (1 equiv, 0.025 M) and **N1** (1 equiv, 0.025 M) in solvent under N$_2$.
[b]Yield collected from chloroform fraction.
[c]Pd(PPh$_3$)$_4$ (5 mol%); yield collected from hexane fraction.
[d]CuTc (2.5 equiv.); yield collected from hexane fraction.

**Table 2 Polycondensation of P1 in classic Stille coupling between E2 and N1[a].**

| Entry | Catalyst | Temperature | $M_n/M_w$ (kDa) | Đ | Yield[b](%) |
|---|---|---|---|---|---|
| 1 | Pd(PPh$_3$)$_4$ | R.T. | - | - | 0 |
| 2 | Pd(PPh$_3$)$_4$ | 100 °C | 11.1/46.1 | 4.17 | 82 |
| 3[c] | Pd(PPh$_3$)$_4$, CuTc | 100 °C | 7.5/22.6 | 3.04 | 61 |

Reaction conditions: [a]Pd catalyst (10 mol%), **E1** (1 equiv, 0.025 M) and **N1** (1 equiv, 0.025 M) in solvent under N$_2$ at room temperature (entry 1) or 100 °C (entry 2).
[b]Yield collected from chloroform fraction.
[c]CuTc (5 equiv.), 100 °C.

polymerization process, the solvent is expected to affect both the catalyst stability and product molecular mass[28,29]. The reaction solvent chlorobenzene (CB) affords polymers with a slightly higher $M_n$ values than toluene (entry 3), which may be ascribed to the better product solubility in the former. Next the more environmentally acceptable solvents, THF, dioxane, and DMF were also screened (entries 4–6), however the polymerization efficiency falls dramatically in all three solvents. Finally, reaction solution concentration effects were scrutinized. Fine tuning the monomer concentrations to 0.1 and 0.001 M (Supplementary Table 1, entries 7–8) yields polymers with $M_n$ below 10 kDa, suggesting the baseline 0.025 M conditions are near optimal in terms of efficiency and molecular mass (Table 1, entry 3). To verify batch-to-batch reproducibility, four parallel trials were performed under the optimized conditions. The product **P1** exhibits nearly identical $M_n$ over 15.0 kDa and comparable yields over 80% (Supplementary Table 2). Additionally, a 1.0 gram scale reaction affords **P1** with $M_n/M_w$ of 15.2/58.6 kDa and a yield of

85% in chloroform fraction (Supplementary Table 2). Therefore, this protocol is quite robust, excellent reproducibility and scalability without obvious fluctuation in $M_n$ and yield.

To provide a direct comparison between classic Stille and the present C-S activation based cross-coupling polymerizations, three parallel reactions were carried out in coupling 2,6-diiodobenzo[1,2-$d$:4,5-$d$']bis(thiazole) (**E2**) and **N1** under various conditions (Table 2). The reaction of entry 1 maintains a colorless solution over 3 days, suggesting negligible activity in a classic Stille polymerization at 25 °C. On increasing the temperature to 100 °C, the reaction affords a polymer with a $M_n$ = 11.1 kDa and Đ = 4.17 after 3 days (Table 2 entry 2), arguing that classic high temperature Stille is less efficient than the 25 °C C-S activation based polycondensation. Interestingly, addition of the CuTc cocatalyst to the Stille system gives inferior activity versus the classic Stille methodology (Table 2 entry 3). Therefore, present copolymerization approach via C-S activation is superior to classic Stille cross-coupling in this system. Moreover, the polymer

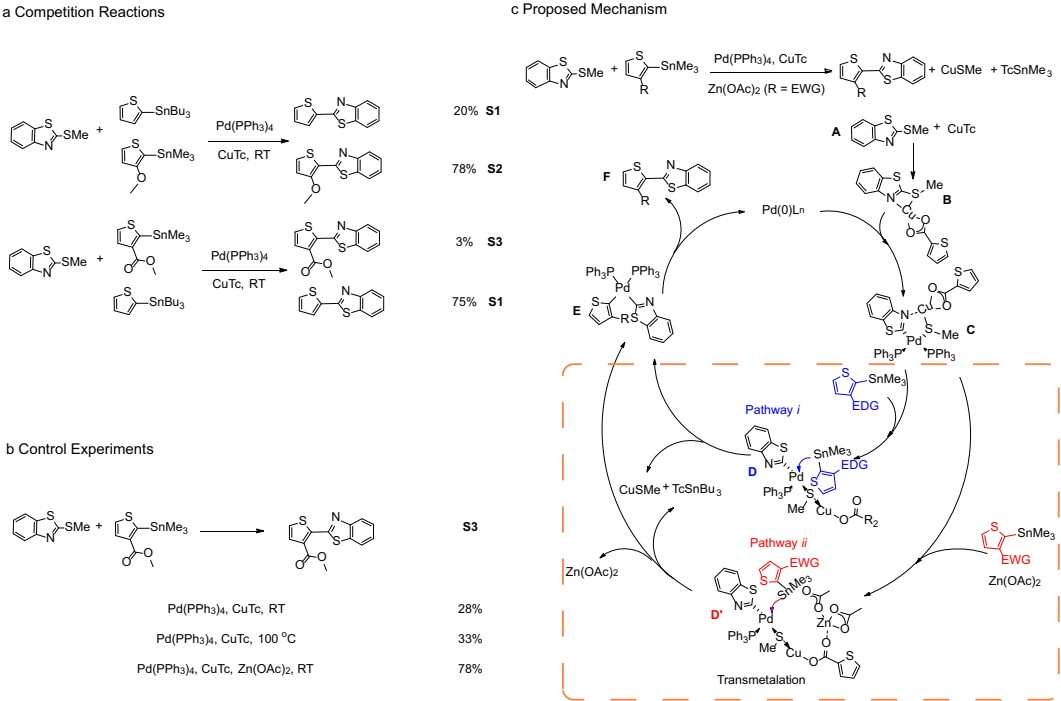

**Fig. 2 Mechanism studies. a** Competition reactions and **b** Control experiment studies of aryl stannane activity and selectivity. **c** Plausible mechanism for C-S cleavage-based cross-coupling polymerization. All yields were determined by gas chromatography (GC) analysis using anthracene as internal standard.

[1]H NMR of the high temperature Stille and the present 25 °C C-S activation products are indistinguishable, with comparable line widths (Supplementary Fig. 1). Additionally, the MALDI-TOF of **P1** (Supplementary Fig. 2) exhibits the expected repeat units corresponding to C-S/C-Sn cross-coupling products ($m/z = 970$), and the end group analysis is consistent with the presence of both monothioether/H and dithioether termini.

**Model reactions and mechanistic considerations**. Small molecule reactions can aid in understanding the utility of a synthetic polymerization method. Note that classic Stille coupling proceeds via a step-growth polymerization mechanism[26], and in this scenario high-yielding reactions are essential for producing high $M_w$ polymers, which is critical to the regularity and electronic properties of the resulting conjugated polymers[30]. To probe the electronic effects of the aryl stannane, competition reactions were performed (Fig. 2a). The direct competition reactions between MTBT, 2-(trimethylstannyl)thiophene, and 2-(trimethylstannyl) thiophene-3-methoxy affords 2-(thiophen-2-yl)benzo[d]thiazole (**S1**) and 2-(3-methoxythiophen-2-yl)benzo[d]thiazole (**S2**) with GC yields of 20% and 78%, respectively. In contrast, competition between MTBT, 2-(trimethylstannyl)thiophene, and 2-(trimethylstannyl)thiophene-3-carboxylate led to GC yields of 75% and 3% for **S1** and methyl-2-(benzo[d]thiazol-2-yl)thiophene-3-carboxylate (**S3**), respectively. These results argue that the more electron-rich stannane nucleophile is more reactive, and that transmetalation is reasonably the rate-determining step in cross-coupling. To investigate the influence of the electron-poor nucleophiles, control experiments in the model reaction between 2-(trimethylstannyl)thiophene-3-carboxylate and MTBT were next conducted. The results reveal that only by heating is the yield increased to 33% (Fig. 2b). Interestingly, addition of Zn(OAc)$_2$ dramatically enhances the yield to *ca.* 80%, consistent with literature reports on related reactions[31]. Based on above studies, a tentative mechanism (Fig. 2c) is proposed, highlighting

the importance of the transmetalation step. It was hypothesized that the C-S bond was activated by CuTc which then facilitates the subsequent Pd(0) oxidative addition[18]. The resultant Pd(II) complex (C) undergoes a transmetalation with electron-rich 2-(trimethylstannyl)thiophene in Pathway *i*. For the transmetalation with an electron-poor 2-(trimethylstannyl)thiophene-3-carboxylate, Zn(OAc)$_2$ plays a critical role in assisting transmetalation in Pathway *ii*. Finally, both pathways involve reductive elimination to afford the target diaryls.

**Reaction scope**. The generality of this catalytic protocol was examined by investigating the coupling of electrophiles **E1-E4** with a series of nucleophiles (**N2-N9**) to afford the corresponding conjugated copolymers **P2-P11**. Table 3 summarizes all polymerizations under the optimized reaction conditions. As electron-rich and rigid ladder-type moieties[32,33], IDT and IDTy distannanes (**N2** and **N3**) were copolymerized with **E1** to smoothly afford copolymers **P2** and **P3** in good yields, respectively. Specifically, **P2** achieves a moderate $M_n$ of 13.4 kDa in a high yield of 82%, mainly due to the strong electron-donating ability of **N2**. Meanwhile, reflecting the introduction of branched alkyl chains, the molecular weight of **P3** is further increased, affording a 22% yield fraction in CB (34.7 kDa). Copolymer **P4** was reported to have the highest hole mobility for a BBT-based polymer, when prepared by classic Stille coupling with a $M_n$ of 26 kDa[34]. In the present work, **P4** was prepared via cross-coupling of **E1** and **N4**, yielding a higher $M_n$ of 30.5 kDa in 60% yield. Note, however, that this reaction yields *ca.* 30% insoluble residue, implying that polymer chain growth is constrained by solubility. Next, the sidechain dimensions of the terthiophenes were increased to *n*-octyldodecyl. Not surprisingly, copolymer **P5** achieves an overall yield of 83% with $M_n = 45.0$ kDa. Furthermore, two more electron-rich units **N6** and **N7** were copolymerized with **E1** to afford polymers **P6** and **P7**, respectively, with moderate $M_n$s and good yields. Since electron-withdrawing moieties are poor nucleophiles in this protocol, nucleophilic DPP distannanes (**N8**)

**Table 3 General applicability of C-S cleavage protocol to other π-electron polymers.**

| entry[a] | Structure | Product | Yield (%)[b] | $M_n/M_w$ (kDa) | $Đ$ |
|---|---|---|---|---|---|
| 1 | | P1 | 84 (CF) | 15.7/49.5 | 3.15 |
| 2 | | P2 | 82 (CF) | 13.4/44.2 | 3.29 |
| 3 | | P3 | 70 (CF) | 16.4/64.6 | 3.94 |
| | | | 22 (CB) | 34.7/109.2 | 3.15 |
| 4 | | P4 | 60 (CB) | 30.5/64.8 | 2.13 |
| 5 | | P5 | 15 (CF) | 13.1/52.2 | 3.97 |
| | | | 83 (CB) | 45.0/150.9 | 3.35 |
| 6 | | P6 | 59 (CB) | 18.5/44.9 | 2.43 |
| 7 | | P7 | 60 (CF) | 9.2/15.4 | 1.67 |
| | | | 17 (CB) | 12.9/21.6 | 1.68 |
| 8[c] | | P8 | 91 (CF) | 11.9/53.6 | 4.52 |
| 9 | | P9 | 18 (CF) | 12.6/25.4 | 2.01 |
| | | | 74 (CB) | 29.7/55.0 | 1.85 |
| 10 | | P10 | 89 (CF) | 17.2/28.8 | 1.67 |
| 11 | | P11 | 64 (CF) | 9.7/16.4 | 1.69 |

Reaction conditions: [a]Pd(PPh₃)₄ (10 mol%), CuTc (5 equiv.), monomers in toluene (0.025 M) for 24 h under N₂.
[b]Yield from chloroform (CF) fraction and chlorobenzene (CB) fraction.
[c]Pd(PPh₃)₄ (10 mol%), CuTc (5 equiv.), Zn(OAc)₂ (10 mol%), monomers in toluene (0.025 M) for 72 h under N₂.

could not successfully generate polymer **P8** in 3 days under the above optimal conditions. Building on Scheme 2b, $Zn(OAc)_2$ was next used as an additive in this polymerization. Copolymer **P8** is obtained with $M_n/M_w = 11.9/53.6$ kDa in 91% yield. Next, arene thioethers **E2**, **E3**, and **E4** were screened in reactions with nucleophiles **N9** and **N5** to afford **P9**, **P10**, and **P11** with $M_n$s ranging from 29.7 to 9.7 kDa, respectively, in good yields, suggesting broad generality of electrophilic thioethers. Finally, 2,5-bis(methylsulfanyl)thiophene and 1,4-bis(methylsulfanyl)benzene were screened as electrophiles. However, no polymers were obtained in the reactions, which may be ascribed to the absence of N atom in the electrophiles that can coordinate with CuTc to facilitate the C-S bond activation. The polymer structures were confirmed with [1]H NMR, high temperature GPC, and elemental analysis. In all, this synthetic methodology is effective for both electrophiles and electron-rich and electron-deficient nucleophiles, suggesting broad future applications.

The photophysical properties of the present **P1-P11** polymers in $CHCl_3$ solution and as thin films were investigated by UV-vis and photoluminescence spectroscopy. The corresponding spectra are shown in Supplementary Figs. 4 and 5, respectively, and the data are summarized in Supplementary Table 3. All polymers in solution exhibit a single major absorption band, which corresponds to intramolecular charge transfer (ICT) excitations at 300–750 nm in solution. The corresponding thin film absorptions for polymers **P1-P3**, **P7-P8**, and **P11** exhibit slight redshifts due to modest π-π stacking[35]. In comparison, **P4-P6** and **P9-P10** thin films show large redshifts of over 21 nm versus their solution spectra, indicating strong interchain π-π stacking in the solid state. The individual photoluminescence spectra of all polymers are shown in Supplementary Fig. 5. **P1-P7** and **P9-P11** solutions in $CHCl_3$ fluoresce with quantum yields of 1.4, 35.6, 26.4, 14.8, 7.2, 5.0, 1.3, 1.0, 6.2, and 1.5%, respectively, with Stokes shifts ranging from 2 to 115 nm. Interestingly, **P8** exhibits no emission, which may be ascribed to strong donor - acceptor unit[35]. Cyclic voltammetry (CV) with the ferrocene/ferrocenium (Fc/Fc⁺) redox couple as an external standard was carried out to characterize the redox properties of all polymers (Supplementary Fig. 6; Supplementary Table 3). The lowest unoccupied molecular orbital (LUMO) and highest occupied molecular orbital (HOMO) energy levels of **P1-P11** are estimated to lie at $-2.87/-5.42$, $-2.80/-5.44$, $-2.96/-5.58$, $-2.81/-5.37$, $-2.73/-5.28$, $-2.94/-5.50$, $-2.99/-5.13$, $-3.62/-5.25$, $-3.50/-5.55$, $-2.80/-5.37$, and $-3.56/-5.68$ eV, respectively. The relatively high-lying HOMO levels of **P1-P7** and **P9-P11** suggest potential p-type semiconductors, while the low-lying LUMO of **P8** suggest possible n-type electron transporting properties (see more below). The decomposition temperatures characterized by thermogravimetric analysis (TGA) of **P1-P11** are all above 300 °C (Supplementary Fig. 7 and Supplementary Table 3), suggesting excellent thermal stability.

**Defect studies**. Homocoupling between electrophiles or nucleophiles are a major deleterious side reaction of cross-coupling reactions, which may cause structural defects in the polymer backbone[8]. To investigate the importance of such side reactions in cross-coupling via L-S type reactions and classic Stille reactions, four parallel experiments were performed (Fig. 3a). Interestingly, in the L-S type reaction, trace amounts of homocoupling product **S5** between nucleophiles but no homocoupling product **S4** between electrophiles are observed in model reactions at 100 °C, while homocoupling products **S5** and **S4** were not observed at room temperature by GC analysis. In comparison, the yield of homocoupling product **S4** is significant (27%) at 100 °C in the classic Stille coupling between 2-(tributylstannyl)thiophene

and 2-iodobenzo[d]thiazole. Not surprisingly, a significant amount of homocoupling product of electrophile **S4** (4%) is still observed for the classic Stille coupling at room temperature. This direct comparison argues that cross-coupling via the present C-S cleavage is plausibly a superior methodology to suppress homocoupling defects in alternating conjugated polymer via catalytic synthesis.

Since **P2** (also refer to **P2-CS**) exhibited sharp peaks in aromatic region of [1]H NMR which is facile to identify the potential homocoupling structural defects. Next, the resulting polymers **P2-CS** (also refer to **P2**) and **P2-CI** were synthesized by our room temperature C-S cleavage method and classic thermal Stille coupling, respectively, to have a direct comparison on structural defects, trap density and further organic field effect transistors (OFET) applications. The MALDI-TOF of **P2-LS** (Supplementary Fig. 3) exhibit the expected repeat units corresponding to C-S/C-Sn cross-coupling products ($m/z = 1094$) indicating the alternating pattern of the polymer backbone. Afterwards, by comparing the aromatic

region of the [1]H NMR spectra of **P2-CS** and **P2-CI** (Figs. 3b, c), the integration of additional proton peak **a'** for **P2-CI** are more than that of **P2-CS**, which may be from the potential homocoupling defects. To further identify the defects ratio of **P2-CS** and **P2-CI**, **PBBT** and **PIDT** were synthesized via thermal Stille coupling. After the careful assignment of **PBBT** and **PIDT** (Supplementary Fig. 3, Figs. 3b, c) chemical shifts, proton **a'** and **a"** of **PBBT** can be assigned at 8.5 and 7.9 ppm which could be applied to identify the BBT homocoupling (hc) ratios of **P2-CS** and **P2-CI**. Basically, the content of BBT homocoupling junctions of **P2-CS** is found to be less than 7%, consistent with the results of the model reactions. In comparison, the homocoupling defects are more than 19% in **P2-CI** (Fig. 3c). This direct comparison underscores the attraction of lower temperature cross-coupling polymerization via C-S cleavage. Although several studies argue that the charge transport mobilities of copolymer is robust against homocoupling structural defects in the OFETs, a deep understanding and interpretation from the trap densities perspective is necessary[36,37]. Here, thermal admittance spectroscopy (TAS) analysis[38] was used to quantize the reduction of trap states in **P2-CS** and **P2-CI** films, respectively. As shown in Fig. 3d, the energetic profile of trap density of states (tDOS) demonstrated a density of defects states on the order of $10^{17}$ to $10^{19}$ for **P2-LS** film, which is relatively smaller than that of **P2-CI** for both shallow and deep defects in the films. Next, OFETs were fabricated with polymers **P2-CS** and **P2-CI** for comparison of the charge transport. As shown in Fig. 3e, f, Supplementary Fig 9, and Supplementary Table 4, the **P2** based OFETs exhibit a good hole mobility of 0.12 cm²/Vs with a high $I_{on/off}$ (on-to-off current ratio) of ca. $10^5$. While, **P2-CI** OFETs have a lower hole mobility of 0.06 cm²/Vs with $I_{on/off}$ of ca. $10^6$, which may reflect the less ordered crystallinity (Supplementary Fig. 10–12) and larger density defects. Again, the results argue that C-S activation based cross-coupling polymerization at room temperature can produce conjugated polymers with comparable or superior microstructural and electronic properties to those from classic Stille coupling at high temperature.

In addition, nanoparticles of **P2-LS** copolymer exhibit excellent co-localization coefficients of 0.79 with lysosome, highlighting the potential application of BBT-based polymers for cell imaging (Supplementary Fig. 13–17). Next, polymer **P4** and **P5** exhibit excellent field-effect transistor hole mobilities (Supplementary Fig. 18–20), rivaling or exceeding literature metrics for the same or comparable materials, further demonstrating the high quality of resulting polymers[34] (Detail discussions are in Supplementary Information).

In summary, we have established a broadly applicable room temperature aryl disulfide C-S polycondensation methodology for

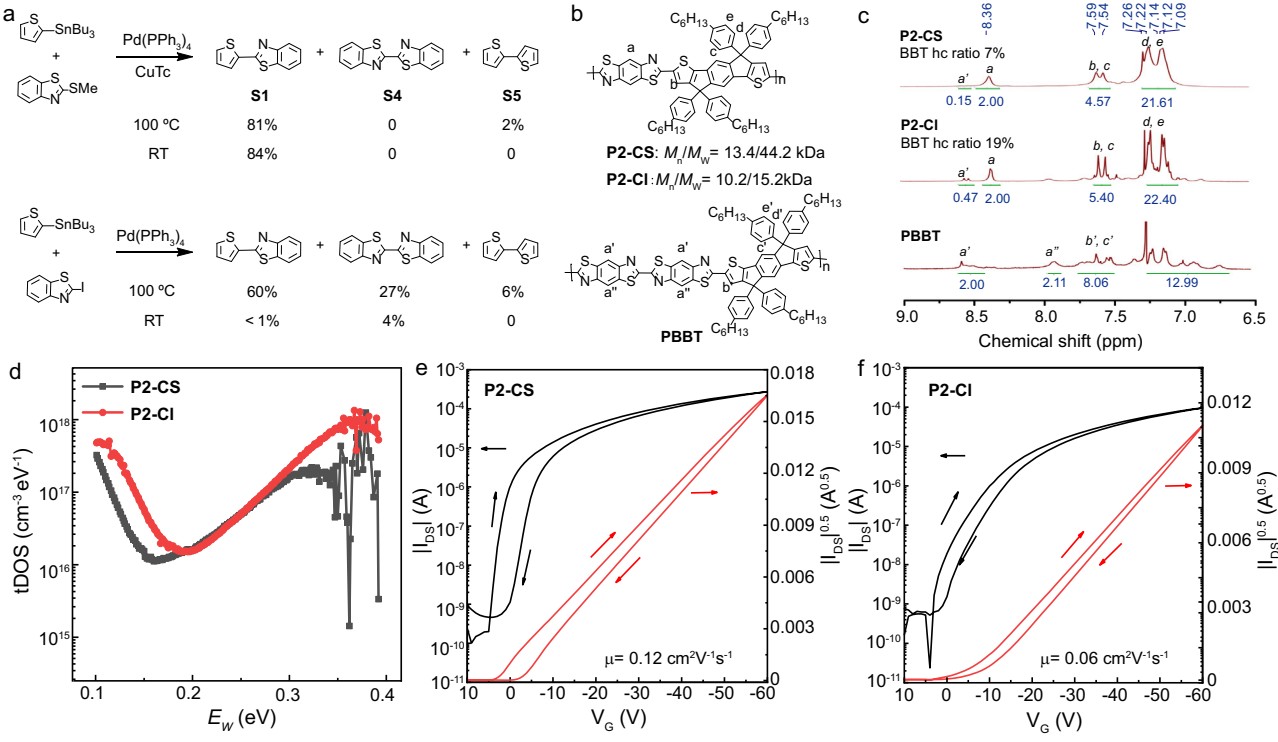

**Fig. 3 Defects studies. a** Side reaction study on generation of structural defects. All yields were determined by gas chromatography (GC) analysis using anthracene as internal standard; **b** The structures of **P2-CS**, **P2-CI**, and **PBBT** synthesized by room temperature Stille via C-S cleavage, thermal Stille via C-I cleavage and thermal Stille, respectively; **c** The aromatic region of $^1$H NMR for **P2-CS**, **P2-CI**, and **PBBT** from 9.0 to 6.5 ppm; **d** Trap density of states (tDOS) obtained by thermal admittance spectroscopy of **P2-CS** and **P2-CI** thin films; **e** OFET transfer curves of polymer **P2-CS** thin film; **f** OFET transfer curves of polymer **P2-CI** thin film.

the efficient cross-coupling polymerization between diverse aryl distannanes and aryl bis-thioethers to produce defect-minimized, high molecular mass alternating semiconducting conjugated copolymers. In comparison to polymer counterparts synthesized via classic Stille polycondensation, the present protocol produces higher molecular masses and defect-reduced polymers in room temperature polymerizations. Detailed mechanistic studies with competition reactions, control experiments, and side reaction analyses provide further guidelines for optimizing on polymer synthesis and minimizing defect generation. A series of alternating polymers was synthesized and shown to have suitable energy levels for semiconduction. It is noteworthy that this structural defect increases the defect density in the bulk films which results in a low OFET hole mobility. Therefore, this polycondensation will not only provide a robust and competitive method for polymer synthesis but also help minimizing the structure defects in high performance polymer semiconductors.

## Methods
**Procedure for polymerizations**. Into a Schlenk flask, **E1-E4** (1 equiv), **N1-N9** (1 equiv), CuTc (5 equiv), Pd(PPh₃)₄ (10 mol%) and dry toluene (0.025 M) were added under a nitrogen atmosphere. The reaction mixture was stirred at room temperature for 24 h, or with 10 mol% of Zn(OAc)₂ as an additive for 3 days. Afterwards, the reaction mixture was poured into methanol. The crude polymer was collected by filtration and purified by successive Soxhlet extractions with acetone, hexane, chloroform and chlorobenzene. The chloroform (CF) or chlorobenzene (CB) solution was then poured into methanol and collected by filtration to afford corresponding product copolymers.

**OFET fabrication and characterization**. A heavily doped Si wafer with a 300 nm thermally grown SiO₂ layer served as gate and dielectric layer, respectively. Before the deposition of the polymers (**P2-CS** and **P2-CI**), the substrates were treated with octadecyltrichlorosilane (OTS) in a vacuum oven at a temperature of 120 °C, forming an OTS self-assembled monolayer. After rinsing by hexane, ethanol and chloroform, 10 mg/mL polymer solution in chlorobenzene was spun coated with a

speed of 5000 rpm, subsequently annealed at 180 °C for 60 min. On top of the polymer thin films, Au source-drain electrodes (30 nm) were thermally evaporated, where the channel length and channel width are 50 μm and 5000 μm, respectively. The OFETs measurements were carried out in ambient using Keysight B2900A analyzer. The mobility in saturation region was calculated with the equation,
$\mu = \frac{2L}{WC_{ox}}(\frac{d\sqrt{I_{SD}}}{dV_G})^2$, where $I_{SD}$ is the source-drain current, $V_G$ corresponds to gate voltage, L and W represent the channel length and channel width, respectively, $C_{ox}$ means the capacitance the SiO₂ layer which equals to 11 nF/cm².

## Data availability
The authors declare that the all the data supporting the finding of this study are available within this article and its Supplementary Information files and are available from the corresponding author on reasonable request.

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

## Acknowledgements

The authors acknowledge the financial support from the NSFC (21774130, 51925306, and 21905277), National Key R&D Program of China (2018FYA 0305800), Key Research Program of the Chinese Academy of Sciences (XDPB08-2), the Strategic Priority Research Program of Chinese Academy of Sciences (XDB28000000), and Fundamental Research Funds for the Central University. T.J.M. thanks the National Science Foundation Materials Research Science and Engineering Center at Northwestern University (DMR-1720319) for support.

## Author contributions

H.H. conceived the work. Z.L. and Q.S. performed the synthetic experiments and characterization and contributed equally to this work. Y. Li and Y. Lin contributed to trap density measurement and analysis. L.Q., X.M. and F.Z. performed the device fabrication and analysis. K.W. performed the cell imaging. H.C. performed the photoluminescence test. W.H. performed the GIWAXS test. Z.L., Q.S., T.J.M., and H.H. wrote the manuscript. H.H. provided overall supervision.

## Competing interests

The authors declare no competing interests.
