## [Peer Review File · Nature Communications]

Efficient Room Temperature Catalytic Synthesis of Alternating Conjugated Copolymers via C-S Bond ActivationREVIEWER COMMENTS

Reviewer #1 (Remarks to the Author):

The authors present a modification of the STILLE cross coupling reaction by using thioether leaving groups instead of halides.

The results demonstrate that this variant can compete with the established standard procedures. However, a couple of additional data is required prior to a final evaluation/decision.

- 1) Please add the ^{13}C NMR spectra of the polymers (to the SI).
- 2) The integration of the ^1H NMR spectra of the polymers (see SI) is very often arbitrary and does not make any sense. Please add the correct integration ranges/limits or skip the integrations.
- 3) Figure 3c is without a ppm scale.
- 4) Please add the PLQs of the polymers. End groups may significantly influence the non-radiative deactivation pathways of the excited states. Especially, please compare the PLQs of P2-CI and P2-CS as similar polymers that are made in different routes.
- 5) Please add the elemental analyses of all polymers. Especially the S values of the polymers may indicate, if additional sulfur is incorporated when using the thioether monomers. Finally please compare the EA of the polymers that result from both coupling methods (e.g. P2-CI and P2-CS)

Reviewer #2 (Remarks to the Author):

This manuscript describes a Pd-catalyzed Stille-type polycondensation reaction by using arylthioethers as electrophilic cross-coupling partners (monomers).

Based on the Liebeskind's condition (CuTC), the authors applied a Pd/Cu catalytic system for the current reaction involving C-S bond cleavage. This combined catalyst was found to be very efficient to access conjugated polyaromatics containing (poly)thiophene skeletons. Reactions by this method can be performed under mild conditions (at room temperature), using commercially available $\text{Pd}(\text{PPh}_3)_4$ as the catalyst and CuTC as the co-catalyst, providing corresponding polymers in good to high yields with high molecular weights. This might lead to a new and practically useful approach for preparations of polyaromatic functional materials. This result also indicates new synthetic utility of arylthioether building blocks. Furthermore, mechanism of this polymerization reaction is also carefully discussed. In general, this manuscript might be suitable for publication in Nature Communications.

Some minor issues:

- 1) For Table 2, what is the result when polymerization reaction between E2 and N1 was performed by catalysis of both $\text{Pd}(\text{PPh}_3)_4$ and CuTC under room temperature?
- 2) In mechanistic study, it shows that addition of $\text{Zn}(\text{OAc})_2$ can efficiently facilitate the cross-coupling in Figure 2b as well as in the case of P8. How about the other polymerizations in the presence of zinc acetate? For example, could the yields/MW of P4, P6, P7, or P11 be further enhanced by adding $\text{Zn}(\text{OAc})_2$?
- 3) The position 2 of thiazole is known of high reactivity. How about the polycondensations of other aryl/heteroaryl thioethers?

Reviewer #3 (Remarks to the Author):

The manuscript by Huang, Marks, Shi and co-workers describes a cross-coupling polymerization method at 25 °C between arylstannanes and arylthioethers. The novelty is mainly in the practical implications in that the reaction temperature is low, yields and molecular weights are high and homocoupling defects are low. The manuscript is of high interest to the community of synthetic organic materials scientists and I would recommend publication with minor adjustments to the text, which I point out in the following.

However, for the supporting information, I recommend acceptance only with major revisions: The data for the synthesized compounds do not meet the standards of the organic chemist: There are

no melting points; there are no IRs and, more importantly, there is no high-resolution MS. ^{119}Sn NMRs are missing. The NMR images do not need a vertical axis and I expect the ^{13}C NMR is broadband decoupled and should be $^{13}\text{C}\{^1\text{H}\}$ NMR.

The logical flow of the manuscript is easy to follow. The main focus is on the synthetic work, starting with the optimisation of the reaction. The observation that lowering the Pd or the Cu loading reduced Mn, but not the yield has not been elaborated. Can the authors provide an explanation? If the catalyst remained coordinated to the growing chain as is typical for living cross-coupling polymerisation reactions, one would expect the opposite: Longer chains with lower catalyst loading. This does not seem to be the case. If the catalyst diffuses away after every step, perhaps it is the progressively slower diffusion of the growing oligomer or polymer chains that reduces their reactivity compared to the faster diffusing smaller oligomers / polymers. It would be really interesting to measure the diffusion coefficients (perhaps by DOSY after first isolating oligomers with preparative GPC). Alternatively, perhaps one could generate end-capped oligomers of different lengths and measure the reaction kinetics with each other and correlate this to this finding. I do realise that this is a lot of extra work and thus perhaps beyond the scope of this manuscript, but such a study in the future would be highly appreciated.

Figure 1a does not depict the Liebeskind-Srogl reaction as stated in the text

In the context of the optimization, I find Table 1 a bit too long and would transfer some of the results to the electronic supporting information.

In Figure 2, c does not really give any important information as it depicts a cross-coupling cycle that is entirely expected. As there are no crystal structures for B or C (or any of the other intermediates), the complexation of the Cu is speculation, although very plausible. On page 7, it says: "It was initially hypothesized that the C-S-bond was activated by..." Why initially? It does not say in the text that this assumption was disproved.

The table with the scope is very impressive and I congratulate the authors on these beautiful results. It seems that in the Sn-monomer, a nitrogen atom is always present. One would assume that this has to do with the complexation to Cu. Did they perform any reaction with a monomer without an N? (in the simplest case just a benzene ring)? Even if this gives poorer results, this should be added, because it would give some more insight into the role of Cu.

I cannot comment on d-e of Figure 3 as this is outside my expertise. What I did find as I was reading the paper was that the nomenclature for the different polymer samples is a bit hard to digest. In Figure 3c, the x-axis is missing. In Figure 3a, I do not understand how it can be that in the second reaction the amounts of homo-coupled products do not add up as in the reaction above. As these are quite important: They should be done with an internal standard so no isolation is necessary and the integration can be done of the reaction mixture.

The term "sloppier peaks" (page 11) is a bit unscientific.

The table of contents graphic is perhaps not as eye catching as it might be - because of a lot of text.

Signed: A. Staubitz

Dear Editors and Reviewers ,

Thank you for your helpful comments on our manuscript “**Efficient Room Temperature Catalytic Synthesis of Alternating Conjugated Copolymers via C-S Bond Activation**” (NCOMMS-21-34694A) to *Nature Communications* which are very important for improving our manuscript.

In response to your comments, we extensively revised the manuscript and highlighted the changes with yellow colors in the manuscript and answer your comments with a point by point as described below.

Reviewer #1:

RE: The authors present a modification of the STILLE cross coupling reaction by using thioether leaving groups instead of halides.

The results demonstrate that this variant can compete with the established standard procedures. However, a couple of additional data is required prior to a final evaluation/decision.

Thanks for the reviewer’s comments. The additional data now was added in the supporting information for final evaluation.

RE: Please add the ^{13}C NMR spectra of the polymers (to the SI).

Thank you for the suggestion. We measured ^{13}C NMR spectra of the conjugated polymers. However, only ^{13}C NMR spectra of **P2-CS** and **P3** presented all C peaks in aromatic and alkyl region. For the rest of the polymers, the ^{13}C NMR spectra of conjugated polymers showed no peaks after over 24 h acquisition time at 130 °C, which may be ascribed to their low solubility and low signal-to-noise ratio. To further consolidate the structure of polymers, we provided the FT-IR spectra and elemental analysis (including C, H, N, and S elements) for all conjugated polymers. Please see more details in supporting information.

RE: The integration of the ^1H NMR spectra of the polymers (see SI) is very often arbitrary and does not make any sense. Please add the correct integration ranges/limits or skip the integrations.

Thank you for your valuable advices. Now all ^1H NMR of conjugated polymers skipped the integration and presented the chemical shift only, since the integration mostly are not very accurate.

RE: Figure 3c is without a ppm scale.

Thanks for the suggestions. We added the ppm scale bar in Figure 3c.

RE: Please add the PLQYs of the polymers. End groups may significantly influence the non-radiative deactivation pathways of the excited states. Especially, please compare the PLQYs of P2-CI and P2-CS as similar polymers that are made in different routes.

Thank you for the valuable suggestions. We measured the PLQYs of **P2-CI** and **P2-CS** to be 30% and 35%, respectively. The PLQYs is indeed affected by the non-radiative deactivation pathways of the excited states due to different end groups in polymers. The data was added into the supporting information.

RE: Please add the elemental analyses of all polymers. Especially the S values of the polymers may indicate if additional sulfur is incorporated when using the thioether monomers. Finally, please compare the EA of the polymers that result from both coupling methods (e.g. P2-CI and P2-CS)

Thank you for the useful suggestions. The C, H, N, and S elemental analyses of all polymers were added to the supporting information. The results showed that no additional sulfur is incorporated into the conjugated polymers when using the thioether monomers.

The elemental analyses for **P2-CI** and **P2-CS** are summarized in supplementary Table 4 for a direct comparison. Obviously, the composition of C and N elements for **P2-CI** is closer to their theoretical data. However, the composition of H and S elements for **P2-CS** are closer to their theoretical data. Overall, the experimental results of **P1-CI** and **P1-CS** are in consist with the theoretical data.

Table S1. Elemental analysis of **P2-CS** and **P2-CI**.

Material	C	H	N	S
	(%)	(%)	(%)	(%)
P2-calcd.	78.93	6.86	2.56	11.70
P2-CS-found	78.66	7.25	2.09	10.75
P2-CI-found	78.89	7.69	2.22	9.21

Reviewer #2:

RE: This manuscript describes a Pd-catalyzed Stille-type polycondensation reaction by using arylthioethers as electrophilic cross-coupling partners (monomers).

Based on the Liebeskind's condition (CuTC), the authors applied a Pd/Cu catalytic system for the current reaction involving C-S bond cleavage. This combined catalyst was found to be very efficient to access conjugated polyaromatics containing

(poly)thiophene skeletons. Reactions by this method can be performed under mild conditions (at room temperature), using commercially available Pd(PPh₃)₄ as the catalyst and CuTC as the co-catalyst, providing corresponding polymers in good to high yields with high molecular weights. This might lead to a new and practically useful approach for preparations of polyaromatic functional materials. This result also indicates new synthetic utility of arylthioether building blocks. Furthermore, mechanism of this polymerization reaction is also carefully discussed. In general, this manuscript might be suitable for publication in Nature Communications.

Thanks a lot for the very positive comments.

Some minor issues:

RE: For Table 2, what is the result when polymerization reaction between E2 and N1 was performed by catalysis of both Pd(PPh₃)₄ and CuTC under room temperature?

Thank you for this good question. The condition of reaction with both Pd(PPh₃)₄ and CuTC is actually the Entry 3 of Table 1. Under this condition, the polymer was synthesized with $M_n = 15.7$ kDa, $D = 3.15$.

RE: In mechanistic study, it shows that addition of Zn(OAc)₂ can efficiently facilitate the cross-coupling in Figure 2b as well as in the case of P8. How about the other polymerizations in the presence of zinc acetate? For example, could the yields/MW of P4, P6, P7, or P11 be further enhanced by adding Zn(OAc)₂?

Thank you for the comments. We tried to add Zn(OAc)₂ into the synthesis of **P4**, **P6**, **P7**, and **P11**. However, their yields/MW were not enhanced at all. We believe that the addition of Zn(OAc)₂ can efficiently facilitate the cross-coupling in Figure 2b and in the case of **P8**, since the nucleophiles (stannies) are electron deficient arenes. However, the addition of Zn(OAc)₂ can not facilitate the cross-coupling of electron-rich nucleophiles, such as the nucleophiles in the synthesis of **P1**, **P4**, **P6**, **P7**, and **P11**.

RE: The position 2 of thiazole is known of high reactivity. How about the polycondensations of other aryl/heteroaryl thioethers?

Thank you for this good question. We agreed that the position 2 of thiazole has high reactivity, which provided good chance for cross-coupling reactions. We did screen other aryl/heteroaryl thioethers, which showed that in addition to thiazole thioethers, tetrazine thioethers (**E4**) also demonstrated good reactivities when coupling with tin reagents, affording **P11** in modest yield and acceptable M_n . The studies on other aryl/heteroaryl thioethers, such as phenyl thioethers or thiophene thioethers, were investigated. However, the current results (see supporting information) showed that no polymer was successfully synthesized under current optimal conditions. Further

optimizations are under investigation in our group, which will be reported in the future.

Reviewer #3:

RE: The manuscript by Huang, Marks, Shi and co-workers describes a cross-coupling polymerization method at 25 °C between aryldistannanes and aryldithioethers. The novelty is mainly in the practical implications in that the reaction temperature is low, yields and molecular weights are high and homocoupling defects are low. The manuscript is of high interest to the community of synthetic organic materials scientists and I would recommend publication with minor adjustments to the text, which I point out in the following.

Thanks for the reviewer's very positive comments.

RE: However, for the supporting information, I recommend acceptance only with major revisions: The data for the synthesized compounds do not meet the standards of the organic chemist: There are no melting points; there are no IRs and, more importantly, there is no high-resolution MS. ¹¹⁹Sn NMRs are missing. The NMR images do not need a vertical axis and I expect the ¹³C NMR is broadband decoupled and should be ¹³C{¹H} NMR.

Thank you for the advice. For all the new small molecules, now we added FT-IR, HR-MS (ESI), melting points, and ¹¹⁹Sn NMR if necessary. Please see them in the supporting information.

We deleted the vertical axis of the NMR images. Also, the ¹³C NMR is ¹³C{¹H} NMR.

RE: The logical flow of the manuscript is easy to follow. The main focus is on the synthetic work, starting with the optimization of the reaction. The observation that lowering the Pd or the Cu loading reduced Mn, but not the yield has not been elaborated. Can the authors provide an explanation?

Thanks for the reviewer's comments. We recognized that the Mn of the polymers dropped significantly, while the yields only decreased slightly when lowering the Pd or the Cu loading. The reason is because this yield and Mn are for the batch of the polymer collected in hexane during the Soxhlet extraction. Under these reactions' conditions, no product was collected from the chloroform fraction.

RE: If the catalyst remained coordinated to the growing chain as is typical for living cross-coupling polymerisation reactions, one would expect the opposite: Longer chains with lower catalyst loading. This does not seem to be the case. If the catalyst diffuses away after every step, perhaps it is the progressively slower diffusion of the growing oligomer or polymer chains that reduces their reactivity compared to the

faster diffusing smaller oligomers / polymers. It would be really interesting to measure the diffusion coefficients (perhaps by DOSY after first isolating oligomers with preparative GPC). Alternatively, perhaps one could generate end-capped oligomers of different lengths and measure the reaction kinetics with each other and correlate this to this finding. I do realise that this is a lot of extra work and thus perhaps beyond the scope of this manuscript, but such a study in the future would be highly appreciated.

Thank you for the interesting question. Actually, we tried to collect the kinetic plot for M_n of **P1** versus monomer conversion. Unfortunately, the monomers were consumed completely within 5 min since GC can't detect any of the monomer **E1**. The results showed that an oligomer with M_n of 1.6 kDa were achieved, which suggested it is a typical step-wise polymerization.

We agreed that that the living cross-coupling is a new area that deserve to be carefully studied in the future.

RE: Figure 1a does not depict the Liebeskind-Srogl reaction as stated in the text.

Thank you for the comments. We revised Fig 1 as 'The present cross-coupling polymerization via C-S activation (Liebeskind-Srogl reaction)' instead of 'The present cross-coupling polymerization via C-S activation'.

RE: In the context of the optimization, I find Table 1 a bit too long and would transfer some of the results to the electronic supporting information.

Thank you for the good suggestion. We transferred several entries to supplementary Table 1 and keep 10 important entries in Table 1.

RE: In Figure 2, c does not really give any important information as it depicts a cross-coupling cycle that is entirely expected. As there are no crystal structures for B or C (or any of the other intermediates), the complexation of the Cu is speculation, although very plausible. On page 7, it says :” It was initially hypothesized that the C-S-bond was activated by...” Why initially? It does not say in the text that this assumption was disproved.

Thank you for the comments. We did try to capture the intermediate of B or C, but failed. To our knowledge, there was no experimental data to support this Cu complexation. This plausible mechanism was proposed based on Liebeskind seminal work (*J. Am. Chem. Soc.* 2000, 122, 11260–11261). We agreed that the word “initially” was confusing. Thus, we deleted it in the sentence.

RE: The table with the scope is very impressive and I congratulate the authors on these beautiful results. It seems that in the Sn-monomer, a nitrogen atom is always present. One would assume that this has to do with the complexation to Cu. Did they

perform any reaction with a monomer without an N? (in the simplest case just a benzene ring)? Even if this gives poorer results, this should be added, because it would give some more insight into the role of Cu.

Thank you for reviewer's positive comments. The scope of Sn-monomers is broad including a series of electron-rich and electron-deficient arene distannanes (N1-N9). However, this methodology currently had limitations on the arene thioethers. The thiazole and its related derivatives showed excellent reactivity, which may be ascribed to the high reactivity of the 2-position of thiazoles. However, for aryl thioethers without a N atom, such as phenyl thioethers or thiophene thioethers, the current results (see supporting information) showed that no polymer was successfully synthesized under current optimal conditions. We speculated that the reactivity of aryl thioether may be related to the coordination of Cu to N atoms. However, more mechanistic study is on the way, which will be reported in the future.

RE: I cannot comment on d-e of Figure 3 as this is outside my expertise. What I did find as I was reading the paper was that the nomenclature for the different polymer samples is a bit hard to digest.

Thanks for the reviewer's modest and meaningful comments. We agreed that the nomenclatures of polymers are sometimes hard to follow. Therefore, we deleted all nomenclatures and simply using the polymer label, such as P1, P2.....

RE: In Figure 3c, the x-axis is missing.

Thank you. For Figure 3c, the x-axis was added.

RE: In Figure 3a, I do not understand how it can be that in the second reaction the amounts of homo-coupled products do not add up as in the reaction above.

Thank you for the good comments. If I understand the reviewer's question right, the reviewer wants to understand why room temperature reaction afforded less homocoupling products.

According to Thompson's review paper (*J. Poly. Sci. Part A: Poly. Chem.* **2015**, *53*, 135-147), the electrophilic homocoupling and cross coupling are competing reactions under current catalytic system. However, the electrophilic homocoupling reaction generally required a higher activation energy than cross-coupling reaction. Moreover, the Eyring-Polanyi equation showed that the reaction rate of the reactions with high activation energy increased faster at elevated temperature. Additionally, Banwell reported the Ullmann type reaction (refer to electrophilic homocoupling; *Acc. Chem. Res.* **2018**, *51*, 1784-1795) mostly occurred only under forcing conditions (often at high temperature). Therefore, high temperature favors the generation of homocoupling byproduct, while room temperature reaction preferred to produce the cross-coupling compounds.

RE: As these are quite important: They should be done with an internal standard so no isolation is necessary and the integration can be done of the reaction mixture.

Thank you for the important comments. I agreed that internal standard is necessary. Thus, we were presenting GC yields using anthracene as internal standard in all mode reactions such as Fig 2a and 2b, Fig 3a.

RE: The term “sloppier peaks” (page 11) is a bit unscientific.

Thanks for your comments. We revised the sentence ‘the proton peaks of **P2-CI** are sloppier than that of **P2-CS** which may be from the potential homocoupling defects’ with ‘the integration of additional proton peak *a*’ for **P2-CI** are more than that of **P2-CS** which may be from the potential homocoupling defects’.

RE: The table of contents graphic is perhaps not as eye catching as it might be - because of a lot of text.

Thank you. We revised the table of contents with simpler words. Please see below image.

Best wishes,

Prof. Dr. **Hui Huang**

College of Materials Science and Opto-electronic Technology
University of Chinese Academy of Sciences
Huairou District, Beijing 101408, P. R. China

E-mail: huihuang@ucas.ac.cn

Tel: +86-10-6967-1736

REVIEWERS' COMMENTS

Reviewer #1 (Remarks to the Author):

From my point of view, the authors did an excellent job in revising their manuscript. I can now recommend acceptance.

Reviewer #2 (Remarks to the Author):

The authors have well addressed my questions.
I think the current manuscript might be published as it is.

Reviewer #3 (Remarks to the Author):

The comments have been dealt with satisfactorily.